# Identification of Landslides in Mountainous Area with the Combination of SBAS-InSAR and Yolo Model

**DOI:** 10.3390/s22166235

**Published:** 2022-08-19

**Authors:** Haojia Guo, Bangjin Yi, Qianxiang Yao, Peng Gao, Hui Li, Jixing Sun, Cheng Zhong

**Affiliations:** 1Badong National Observation and Research Station of Geohazards, China University of Geosciences, Wuhan 430074, China; 2Three Gorges Research Center for Geo-Hazard, Ministry of Education, China University of Geosciences, Wuhan 430074, China; 3Yunnan Institute of Geological Science, Kunming 650051, China; 4Department of Earth and Ocean Sciences, University of North Carolina, Wilmington, NC 28403, USA or; 5Department of Geography, University of South Carolina, 709 Bull St., Columbia, SC 29208, USA; 6School of Earth Sciences, China University of Geosciences, Wuhan 430074, China

**Keywords:** landslides, Sentinel-1, InSAR, deep learning, high resolution image

## Abstract

Landslides have been frequently occurring in the high mountainous areas in China and poses serious threats to peoples’ lives and property, economic development, and national security. Detecting and monitoring quiescent or active landslides is important for predicting risks and mitigating losses. However, traditional ground survey methods, such as field investigation, GNSS, and total stations, are only suitable for field investigation at a specific site rather than identifying landslides over a large area, as they are expensive, time-consuming, and laborious. In this study, the feasibility of using SBAS-InSAR to detect landslides in the high mountainous areas along the Yunnan Myanmar border was tested first, with fifty-four IW mode Sentinel-1A ascending scenes from 12 January 2019 to 8 December 2020. Next, the Yolo deep-learning model with Gaofen-2 images captured on 5 December 2020 was tested. Finally, the two techniques were combined to achieve better performance, given each of them has intrinsic limitations on landslide detection. The experiment indicated that the combination could improve the match rate between detection results and references, which implied that the performance of landslide detection can be improved with the fusion of time series SAR images and optical images.

## 1. Introduction

The high mountainous areas of China are prone to landslides due to active tectonic movement, complex geological environment, diverse climate types, and intense human activities [1,2]. In 2020, China experienced 4810 landslides, resulting in more than 117 dead or missing, and more than 717 million USD in economic losses. Detecting landslides is the first step in developing strategies of predicting risks and mitigating losses [3]. However, it is still a very difficult task to detect landslides, especially in mountainous areas covered by dense forests [4]. Detecting the surface deformation is a popular way to identify landslides, as it is an apparent indicator of active ones [5]. However, traditional methods, such as the global positioning system (GPS), total station, and inclinometer, are all expensive, time-consuming, and laborious. Thus, they are often used in field investigation at a specific site rather than over a large area [5,6].

As a new geodetic technique, InSAR can detect millimeter-level surface deformation over a large area, and is barely affected by cloud or weather [7,8,9]. Many cutting-edge InSAR technologies have been successfully implemented in detecting slope movement or identifying landslides. Liu et al. [2] successfully detected 17 unstable slopes through developing a coherent scatterer time-series InSAR method. Tomás et al. [10] used wavelet tools to analyze the seasonal variations in the Huangtupo landslide from InSAR time-series data. Lu et al. [11] successfully used persistent scatterers interferometry hotspot and cluster analysis (PSI-HCA) to detected extremely slow-moving landslides. Crosetto et al. [12] successfully detected eight potential landslides in a large mountainous area of the Yalong River Basin. Dai et al. [13] monitored the activity of the Daguangbao mega-landslide in China using Sentinel-1 TOPS time-series interferometry. However, the InSAR processing faces many challenges in high mountainous areas [14], such as incoherence, atmospheric delay, and geometric distortion, caused by steep terrain, dense vegetation, etc. Thus, it is still difficult to achieve accurate and reliable results when detecting landslides in these areas with InSAR. 

Alternatively, many endeavors have been directed toward identifying landslides from optical satellite images, including manual interpretation, change detection, unsupervised or supervised classification, machine learning algorithms, and object-oriented analysis [15,16,17,18]. Recently, the known deep-learning models have been successfully used in landslide detection and achieved outstanding results [19]. The results of comparative tests verified the DCNN model could better determine the mean intersection over union (IoU) than shallow machine learning methods, e.g., the artificial neural network, support vector machines, and random forest [20]. Furthermore, the residual network (ResNet), DenseNet, U-Net model, LandsNet, Mask R-CNN, etc., have been tested in landslide detection [19,21,22,23,24,25,26]. However, landslide detection from optical images is still facing great challenges. First, the spectral and spatial characteristics of bare slopes or ranks are often similar to those of landslides, so they are hard to distinguish. Second, the differences between landslides modeled by different geo-environment factors are often too huge to find a uniform detection criterion. Third, landslides covered by dense vegetation cannot be discovered when solely using spectral information.

In this study, the feasibility of using SBAS-InSAR to detect landslides in the high mountainous areas along the Yunnan Myanmar border was first tested, with fifty-four IW mode Sentinel-1A ascending scenes from 12 January 2019 to 8 December 2020. Second, the Yolo deep-learning model with Gaofen-2 images captured on 5 December 2020 was tested. Finally, the two techniques were combined to achieve better performance, as each of them has intrinsic limitations in landslide detection. The experiment proved that the combination could evidently improve the match rate between detection results and references, which implied it is a promising method to improve the performance of landslide detection through the fusion of time-series SAR images and optical images.

## 2. Study Area

Fugong County in Yunnan Province is located at the southeast of the Qinghai Tibet Plateau, Nujiang Valley, between Biluo Snow Mountain and Gaoligong Mountain, with a land area of 2756.44 square kilometers, as shown in Figure 1. The border between the county and Myanmar is 142.218 km. As of 2020, the county has a total population of 114,372, belonging to 20 ethnic groups. As shown in Figure 1, about 75% of the county was covered by dense forest, while arable land only covered 5%. Additionally, 85% of the arable land was located on steep slopes (>25°). The average yearly precipitation was 1443 mm, which mainly happened in the two rainy seasons, e.g., the spring and early autumn, and was affected by the Indian Ocean and Pacific monsoons.

Fugong County is located in the arc turning part extending south from the Tibetan Plateau of the Gangdese–Nyainqingtangula and Sanjiang fold systems. The general structural direction changes from north–west to north–south. The two first-order tectonic units are divided by the Lancangjiang fault, forming the eastern and western tectonic belts. The neotectonic movement of Fugong County belongs to the tilting uplift area in western Yunnan. According to the intensity of neotectonic movement, it is further divided into the Lanping Simao secondary fault block uplift. The strata of the area mainly include Proterozoic, Upper Paleozoic, Mesozoic, and others. Among them, the upper Paleozoic is most widely distributed. The east Lanping Simao depression area is dominated by Mesozoic strata, while the west Gangdese–Nyainqingtangula fold system area is dominated by upper Paleozoic Carboniferous strata, Proterozoic Gaoligongshan group and Chongshan group metamorphic rocks, and Yanshanian and late Variscan granites. Since 1950, the area has witnessed six great earthquakes, which have significantly influenced the slopes’ stability.

## 3. Methods

### 3.1. Data

#### 3.1.1. Remote Sensing Images

Fifty-four IW mode Sentinel-1A ascending scenes covering the study area and spanning from 12 January 2019 to 8 December 2020 were obtained from https://search.asf.alaska.edu (accessed on 1 June 2021). The wavelength, resolution, and width of Sentinel-1A images are 5.6 cm, 5 × 20 m, and 250 km, respectively. The spatial–temporal baselines of scenes are mapped in Figure 2. In order to reduce the phase error caused by orbit errors, the precise orbit data of the study area obtained from European Space Agency were used to refine the orbit information.

The Gaofen-2 images captured on December, 2020, having 1 m panchromatic and 4 m multispectral (Blue, Green, Red and Near infrared) bands, were collected for identifying geological hazards using deep-learning (DL) methods. The Bijie Landslide Dataset [22] consisting of 770 landslide samples and 2003 no-landslide samples, was collected to train and validate the DL model. The samples were produced from TripleSat satellite multispectral images, which have a spatial resolution of 3.2 m, similar to that of Gaofen-2 images.

#### 3.1.2. References and Ancillary Data

Landslides manually identified through interpreting images and field checking in 2020 were collected from the Yunnan Institute of Geological Science to verify the results from SBAS-InSAR, deep learning, and their combination. There were 338 landslides in the inventory, which were mainly distributed along the Nujiang River. 

In addition, the 30 m resolution SRTM1 digital elevation model (DEM) was downloaded for the removal of topography phase (a phase change induced by topography) in InSAR processing. A 1:50,000 topographic map was collected for analyzing the influences of environmental factors (geology, terrain, human activity, etc.). The 30 m land use map of 2015 [27] was downloaded to analyze the distribution of InSAR scatter points.

### 3.2. Methods

#### 3.2.1. SBAS-InSAR

First, SAR images were grouped into multiple small subsets according to specific thresholds of spatial and temporal baselines Therefore, the baseline distances become shorter in each subset, although the distances between subsets may be long. In this way, the incoherence caused by long baselines could be alleviated. Then, least square method was used to obtain the surface deformation of each subset. Finally, singular value decomposition (SVD) was conducted on the combination of all subsets to detect the time-series deformation during the whole period. In the process, the influences of atmospheric delay and DEM uncertainty must be removed by introducing related external data, to guarantee accuracy [28].

In this study, the SBAS-InSAR was processed in the open-source system GMTSAR [29,30], as shown in Figure 2. Firstly, DEM and precise orbit data were imported to remove the topographic phase (a phase change induced by topography) and refine the orbit information. Then, the spatial and temporal baselines of the SAR images were estimated to obtain the flat phase and clues for selecting interference pairs. Third, appropriate thresholds of spatial and temporal baselines were assigned, which were 61 days and 75 meters, respectively; then 157 interference pairs were selected, which are shown in Figure 3. Fourth, differential interference was implemented on selected pairs to obtain the differential interferogram, followed by the phase unwrapping to determine the integer ambiguity. Finally, LS and SVD were implemented to solve the phase equations of the interferograms and estimate the surface deformation. 

#### 3.2.2. Yolo Model

You only look once (Yolo) is a real-time and end-to-end object detection model [31]. In Yolo, object detection is regarded as a regression issue, and the position and category of the preselection box are directly predicted in the output layer. Only one CNN and a single detection are needed to obtain the category probability and coordinate of the object. The latest version of Yolo is extremely fast and accurate, and is thus very suitable for remote sensing big data interpretation.

As shown in Figure 4, the Yolo model consists of DarkNet for extracting backbone features and feature pyramid networks (FPNs) for merging multiple-scale features. In FPNs, multiple scale feature maps (13 × 13, 26 × 26, and 52 × 52) were produced using the upsample strategy, and then fused for detecting small landslides. The method can simultaneously make full use of the accurate location information in low-level features and rich semantic information in high-level features.

In Yolo, the whole image is divided into s × s grids. The center of a grid is considered the center of the candidate box. Object detection is conducted in the grid where the center of an object falls. The predicted candidate boxes of grids are depicted by five parameters, *x*, *y, w*, *h* and *c*. Here, (*x*, *y*) are the coordinates of the center of predicted boxes; (*w*, *h*) are the width and height of the box, respectively; the confidence *c* indicates the IoU between the predicted box and true box. Finally, the product of the confidence and category probability is used to evaluate the reliability of the prediction.

As the number of samples was limited, augmentation strategies were conducted to enlarge the sample set and avoid the problem of overfitting, which included crop, rotation, flip, mosaic, and blur. The former three can help the Yolo model learn the characteristics of landslides from different angles. Mosaic was used to enrich the features of the background and small objects through mosaicking patches, which were randomly cut, clipped, and arranged. When using Blur, noises were randomly added to samples, and then the robustness of Yolo model could be improved through learning blur images.

## 4. Results

### 4.1. Landslides Detected by SBAS-InSAR

At first, the annual surface deformation rate of Fugong County was obtained using SBAS-InSAR technology, as shown in Figure 5; then, possible landslides were identified according to an empirical threshold and checked with the references collected by visual interpretation.

As shown in Figure 4, few scatter points were detected in most areas of the county that were covered by dense trees, as SBAS-InSAR techniques likely suffer from interference there. Meaningful scatter points were mainly found in valleys along the Nujiang River and its branches. For the 41,032 scatter points, the average deformation rate was 3.8 mm/y with a standard deviation of approximately 9.3 mm/y, while the absolute average value is 11.1 mm/y with a standard deviation of about 5.1 mm/y. Figure 6a indicates 90% of them are located in the range from −20 mm/y to 20 mm/y. Through overlapping scatter points with the land cover map, it is found that 35.24%, 24.33%, 16.96% and 13.17% of them are located on forest, grassland, cultivated land and impervious surface, as shown in Figure 6b. It is also noted that scatter points match reference landslides well, implying the possibility of using SBAS-InSAR results to identify landslides.

From the SBAS-InSAR detection result, possible landslides could be identified with an appropriate threshold of deformation rate [15]. In this study, 10 mm/y was selected as the threshold according to previous studies [9] and landslide’s activity in the study area. Finally, 401 possible landslides were identified and then checked with the reference landslides collected by manual interpretation, as shown in Figure 7.

Figure 7 shows that most references (i.e., landslides identified by manual interpretation) were detected by SBAS-InSAR, as TPs accounted for 68.75% of the total references. In other words, most of them remained active during the study time, and may pose serious threats to their surroundings. Thus, public and local authorities should pay closer attention to them. With regard to those references not detected by SBAS-InSAR, we concluded that they remained inactive during the study period. In addition to the errors or incoherence in SBAS-InSAR processing, the finding that SBAS-InSAR could not detect inactive slopes also led to the mismatch between its results and references. We also note that many possible landslides identified by SBAS-InSAR were not recorded in the reference inventory, suggesting that SBAS-InSAR may provide valuable clues for finding landslides beyond the coverage provided by ground investigation.

### 4.2. Landslide Identification with Yolo

In this study, landslide identification from optical images was implemented with the Yolo model. The training process of Yolo model went through 400 epochs, and the model tended to converge after 280 epochs. Finally, the mAP50 and mAP50:95 reached 99.17% and 73.5%, respectively. Here, mAP50 refers to the detection accuracy when the IoU is greater than 0.5, while mAP50:95 indicates the average detection accuracy when the IoU grows from 0.5 to 0.95 in intervals of 0.05. The results of Yolo are shown in Figure 8, which were checked with the references collected by manual interpretation. 

Figure 8 and Figure 9 illustrate a comparison between the Yolo results and references collected by manual interpretation. Among the 327 possible landslides, merely 65.75% were verified by the ground references. This confirmed that it is difficult to identify landslides in high mountainous areas from optical images [31]. Shallow- or deep-learning machine learning methods can only learn spectral and spatial characteristics from the image. In this case, crop or bare land, which display similar spectral and spatial characteristics to landslides, are easily misclassified. In addition, landslides covered by forests cannot be detected from optical images, as their characteristics are similar to those of forests. Therefore, the accuracy of identifying landslides from optical images is often not satisfied for direct applications. Further field survey is necessary to verify the results.

### 4.3. The Combination of SBAS-InSAR and Yolo

Considering the two cutting-edge techniques have their own characteristics and limitations, we attempted to combine them to check if the performance of landslide detection could be improved. 

Figure 10 illustrates the comparison between the combination results and references collected by manual interpretation. Among the 388 references, 80.41% were identified by the combination and labeled as TP, which is much higher than that attained by either of the two methods. This confirmed that the fusion of time series SAR images and optical images is a promising alternative to improving the performance of landslide detection by fully exploiting their temporal, spatial, and spectral characteristics with cutting-edge techniques. Note that the distribution area of FPs is greater than that of the references, which implies the combination can provide valuable information for finding landslides beyond the areas humans can reach. We found that 42.3% of the Yolo result was also identified by SBAS-InSAR; thus, adding the InSAR result into the DL classification model as a temporal feature may be a feasible way to fuse the two types of images for improving the detection performance in future work. Figure 9 and Figure 11 show examples of landslides identified by Yolo and SBAS-InSAR, and verified by references or field work.

## 5. Discussion

### 5.1. Natural and Anthropogenic Factors

To understand the mechanisms of landslides and evaluate their risks, the natural and anthropological factors affecting their development were analyzed with a DEM, topographic map, and land cover map of the study area.

Figure 12a shows the number of landslides generally decreases when the elevation increases. Specifically, 83% were observed below 2.0 km, while merely 17% were found when the elevation was higher than 2.0 km. This implies that lower slopes are more affected by rainfall, river flows, intense human activities, and engineering constructions. Slope gradient has a significant influence on landslide development, through affecting the runoff of surface water, the recharge and discharge of groundwater, and the transportation and accumulation of soil deposits, etc. Figure 12a shows most landslides occur on moderate slopes (e.g., from 10° to 35°), which is in accordance with the general trend in landslide distribution. The altitude also has significant impacts on slope stability, through affecting the time and intensity of the sun radiation, the distribution of rainfall, and then the erosion on slopes. Figure 12a shows most landslides are located in the west directions (e.g., SW, W, and WE), which may partially be attributed to the poor capacity of Sentinel-1A to observe slopes in the north and south directions.

Rivers can influence the soil moisture of their banks and undercut slope toes, which may directly result in slope movement, especially in rainstorm periods. Figure 12b indicates the number of landslides generally decreases when the distances to rivers increase. Specifically, 32.9% were located close to rivers (a distance of less than 200 m). Frequent engineering constructions and other human activities around roads may cause slope instability and landslides. Figure 12b suggests that 80% of landslides were located close to roads (a distance of less than 200 m), which implies the influences of human activities on slope stability. In high mountainous areas, most roads are built along rivers or valleys,; thus, the simultaneous impacts of human activities and rivers make slopes become more unstable.

In this study, the development of landslides was also seriously affected by poor geological conditions. The county is located in the valleys of Hengduan Mountains, where the Nujiang and Lancangjiang faults are still active after several structural changes, resulting in the development of broken rock mass structures, steep slopes, and dense water systems. In addition, the county is also characterized by vertical zoning climate, uneven distributed precipitation, and strong rainstorms. Due to these adverse conditions, the county is very prone to geological disasters, where landslides often occur, which lead to serious damage. 

### 5.2. Contributions, Limitations, and Further Directions

SBAS-InSAR could accurately detect the surface deformation of distributed targets from time-series SAR images, with multiple small subsets and phase optimization. Compared with PS-InSAR, the coverage and density of its scatter points are greater in mountainous areas. Although about 70% of SBAS-InSAR results matched the ground refences well, the correspondence between them was still not clear, direct, or inevitable. First, InSAR processing often suffers from incoherence, atmospheric delay, and geometric distortion, due to the steep terrain and dense vegetation in high mountainous areas, which likely lead to detection errors. Second, SBAS-InSAR could detect deforming slopes, but as not able to discover quiescent landslide. Thus, the technique is suitable for identifying emerging threats, but is not competent for a comprehensive investigation of potential landslides. Other InSAR techniques can be tested for improving the reliability of landslide identification.

Although many cases of identifying landslides from optical images with shallow- or deep-machine learning algorithms have been reported, it is still difficult to distinguish them from bare land or banks in optical images [32]. First, the spectral and spatial characteristics of landslides are often similar to those of bare slopes, banks, etc. Second, landslide elements (e.g., main scarps, tongues, and bodies) have different spectral and spatial characteristics. Third, those covered by dense forest, grass, and even buildings cannot be identified from this type of image. Additionally, optical satellites images are easily contaminated by haze, cloud, and other obstacles [33]. Using one static optical image cannot measure the surface displacement. With regard to algorithms, the application of DL in landslides is still in its infancy, whose network architectures, training samples, augmentation strategies, and more need further improvement [34].

In this study, we combined SBAS-InSAR and Yolo model, and achieved better performance than either of them separately. Importantly, we found that the coverage of the combination result was much wider than that of the references, suggesting it is a valuable supplementary to ground investigation, especially in remote and dangerous areas. In future research, other methods can be explored to fuse multiple sources images, and exploit the spectral, spatial, and temporal characteristics of landslides. For instance, the InSAR detection results may be fed into the DL classification model as the temporal feature for improving the detection performance.

## 6. Conclusions

In this study, SBAS-InSAR, Yolo model, and their combination were tested for detecting landslides in a high mountainous county. We found that the InSAR technique could only detect deforming slopes, and that it was still difficult to distinguish landslides from bare slopes or banks in optical images using either shallow- or deep-learning methods. Experiments showed that their combination achieved better performance than either of them separately and covered a greater area than field surveys. This study confirms that the combination or fusion of multiple sources of remote sensing images is a promising alternative to improving the performance of landslide detection, through fully exploiting their temporal, spatial and spectral characteristics with cutting-edge techniques. This study provides valuable results and clues for authorities to develop strategies of discovering landslides and mitigating the potential losses they cause in high mountainous areas.

## Figures and Tables

**Figure 1 sensors-22-06235-f001:**
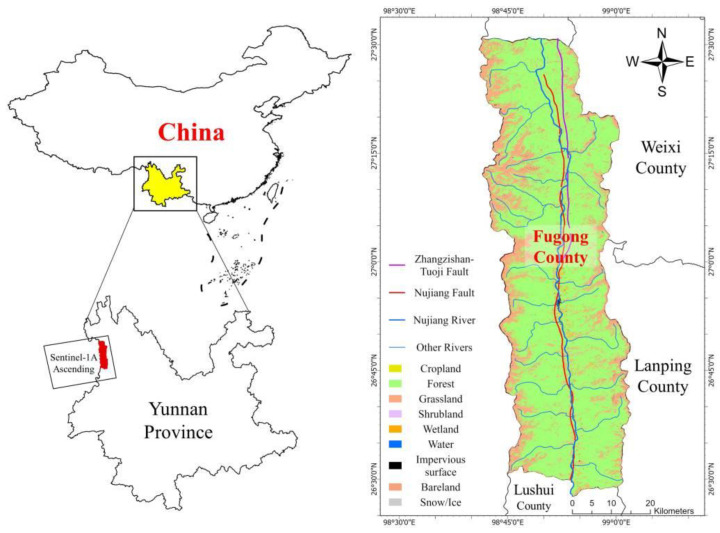
The location and land cover of the study area.

**Figure 2 sensors-22-06235-f002:**
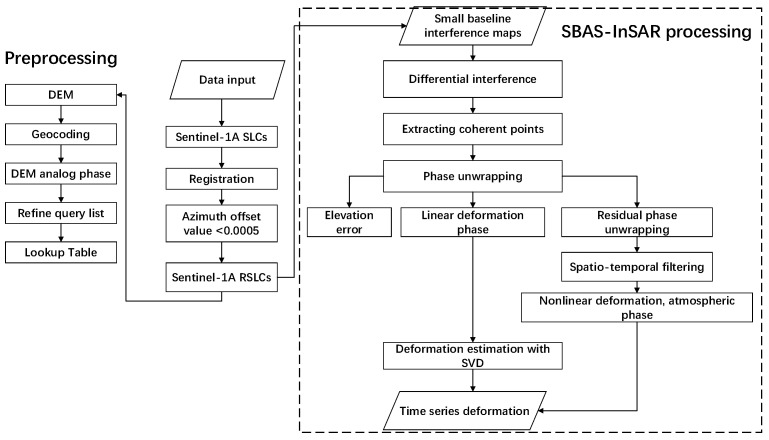
The flowchart of SBAS-InSAR technique.

**Figure 3 sensors-22-06235-f003:**
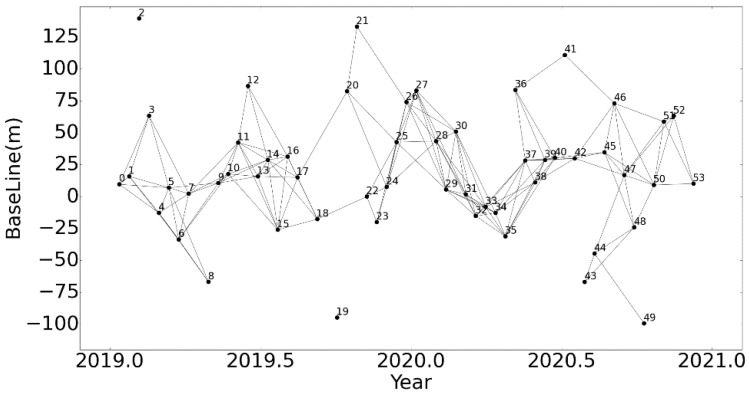
The baseline map of Sentinel-1 images.

**Figure 4 sensors-22-06235-f004:**
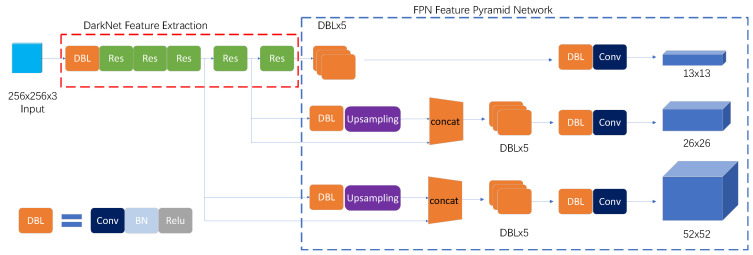
The Yolo model.

**Figure 5 sensors-22-06235-f005:**
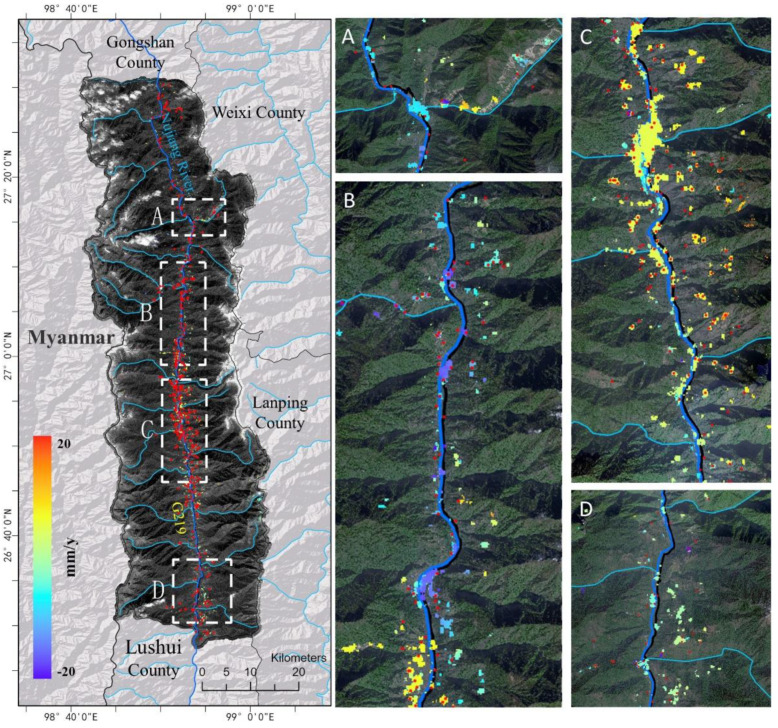
The surface deformation detected by SBAS-InSAR. Here, the color bar in the left panel shows the color of different annual deformation rates in the LOS direction, where a negative value indicates increasing distance from the satellite, and a positive value indicates decreasing distance from the satellite. The red points are references collected by visual interpretation. Panel (**A**–**D**) are used to display specific regions of the left panel more clearly.

**Figure 6 sensors-22-06235-f006:**
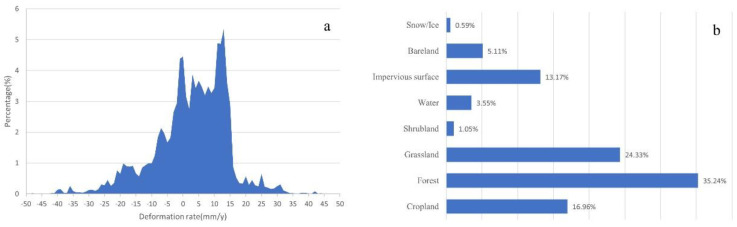
The distribution of scatter points. (**a**) the annual deformation rate of them; (**b**) the land cover of scatter points.

**Figure 7 sensors-22-06235-f007:**
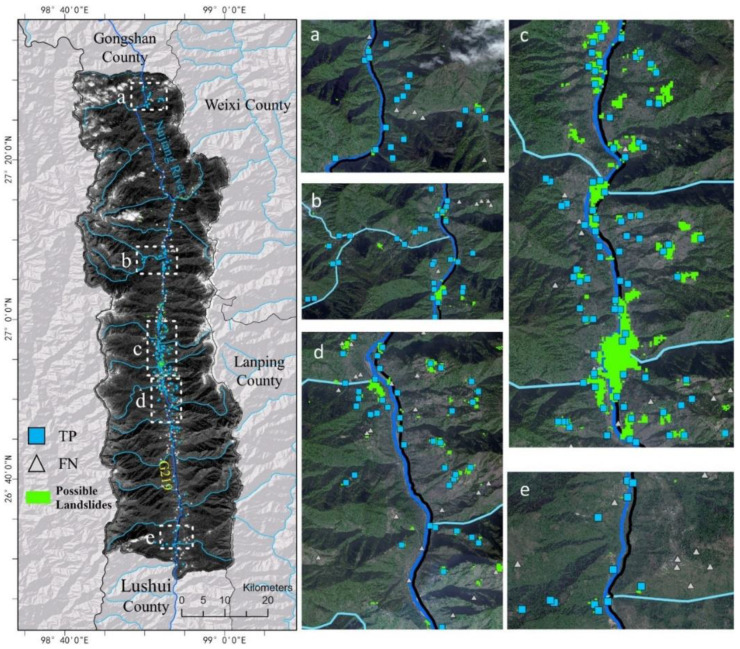
The comparison between possible landslides identified by SBAS-InSAR and references. Here, TP indicates the intersection of SBAS-InSAR landslides and references, while TN indicates the left references not matched with SBAS-InSAR landslides. Here, scatter points having deformation rate > 10 mm/y are considered possible landslides. Panel (**a**–**e**) are used to display specific regions of the left panel more clearly.

**Figure 8 sensors-22-06235-f008:**
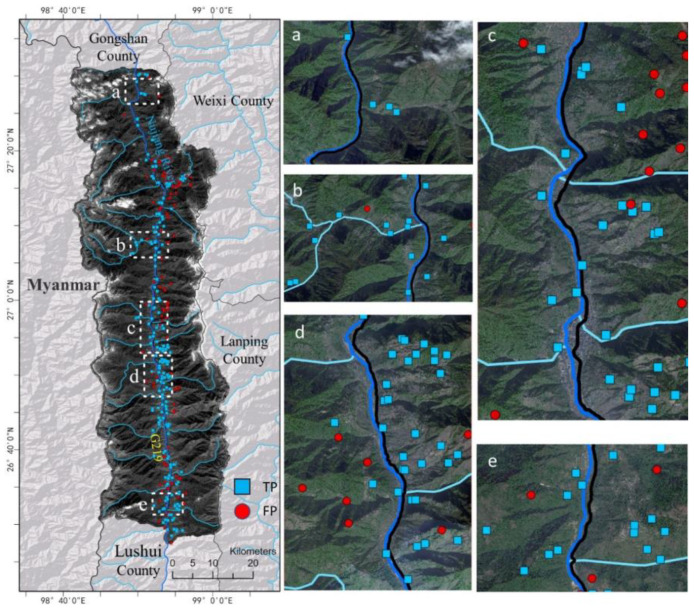
The possible landslides detected by Yolo. Here, TP is the intersection of Yolo results and references, while FP indicates those not recorded in the reference inventory. Panel (**a**–**e**) are used to display specific regions of the left panel more clearly.

**Figure 9 sensors-22-06235-f009:**
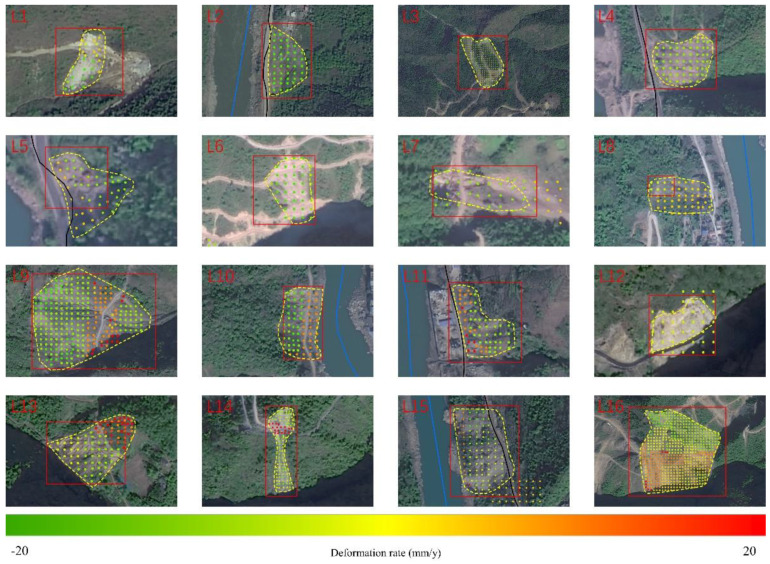
The examples of possible landslides identified by Yolo, which were also detected by SBAS-InSAR and verified by references. Here, the rectangles indicate Yolo results, scatter points represent SBAS results, and while yellow polygons indicate references.

**Figure 10 sensors-22-06235-f010:**
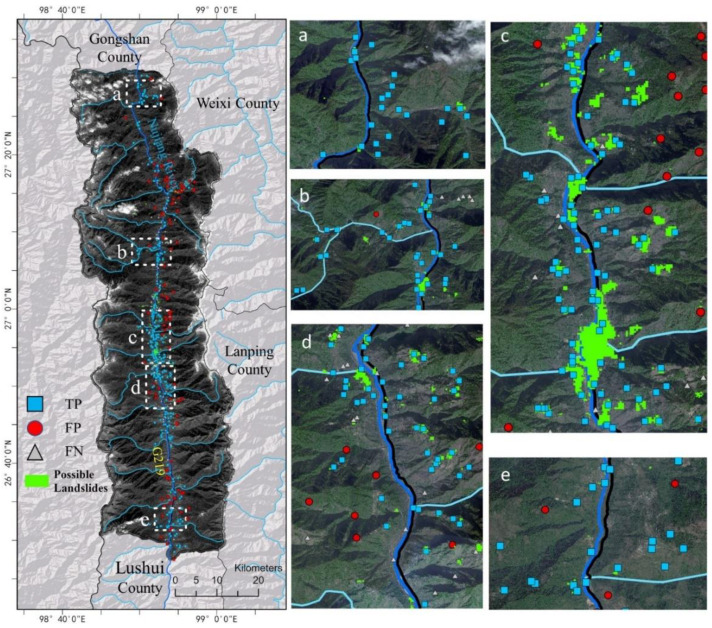
The results of the combination of SBAS-InSAR and Yolo. Here, TP is the intersection of the combination results and the references; FP indicates those not recorded in the reference inventory; and FN indicates references not identified by the combination. Scatter points having deformation rate > 10 mm/y are considered possible landslides. Panel (**a**–**e**) are used to display specific regions of the left panel more clearly.

**Figure 11 sensors-22-06235-f011:**
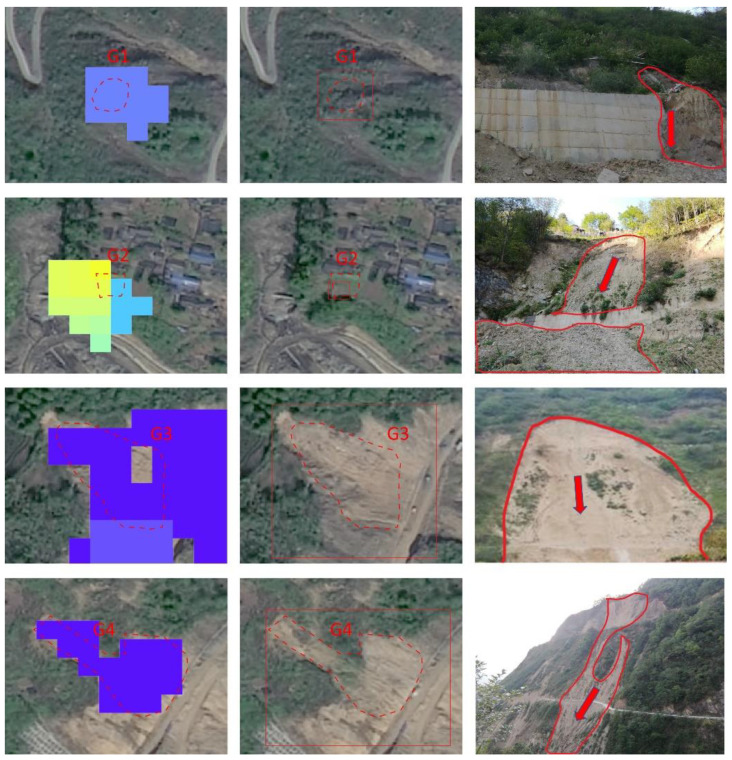
Examples of landslides identified by (from left to right) SBAS-InSAR, Yolo, and verified field work. Here, the color pixels in left panels indicate the scatter points; red rectangles and dotted polygons in middle panels represent the Yolo result and reference, respectively; the polygons in right panels indicate the landslides delineated in field work and the arrow displays the main slide direction.

**Figure 12 sensors-22-06235-f012:**
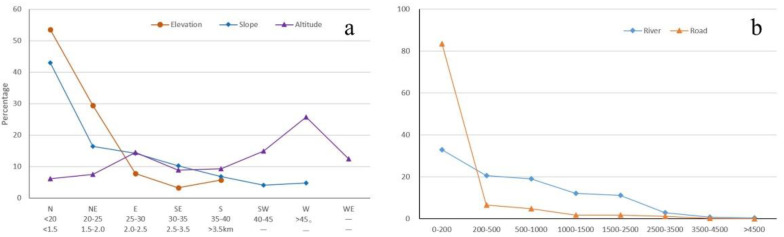
Factors influencing landslide: (**a**) the topographical factors; (**b**) rivers and roads. In panel a, the 1st, 2nd, and 3rd rows of the caption below the lateral axis indicate the category of altitude, slope, and elevation, respectively.

## Data Availability

The data presented in this study are openly available in https://search.asf.alaska.edu (accessed on 1 June 2021).

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
