# Peer review of "Identification of Landslides in Mountainous Area with the Combination of SBAS-InSAR and Yolo Model"

_sensors, 2022, doi:10.3390/s22166235_

Round 1

Reviewer 1 Report

Dear authors,

I reviewed the paper entitled “Identification geological hazards in mountainous area with the combination of the SBAS-InSAR and Yolo model” by Haojia Guo et al. The paper tries to apply a combination of a multi-temporal InSAR analysis and the Yolo classification algorithm to detect and highlight hazards in a mountainous area in China.

After reading the paper there are some things not clear.

First, the paper needs an English revision. There are many grammar errors and typos. Some parts are difficult to read.

Abstract. It should indicate the SAR sensor and the dates of the Gaofen-2 images used in the study. It talks about the SBAS-InSAR method but it does not say anything about PS-InSAR which is used in the paper. Some results should be given.

Introduction. Replace “Literature (X)” with the name of the author/s. It should indicate the methodology and data used in the study, as well as state clearly the objectives of the study. The introduction should also indicate the main results of your study.

Section 2.1. Indicate the Nujiang and Zhangzishan Tuoji faults in Figure 1.

Section 2.2. Remote sensing images. Why did not you use a large Sentinel-1 data set? Why only 2 years if you have data from more years available? How many Gaofen-2 images did you use in the study? Which dates? Place the footprint over Figure 1.

In Figure 2, why are there isolated images not linked to the rest?

In Section 3.1 you talk about different subsets. Which subsets do you refer to? Apparently, there is only one subset. Which reference did you use in the SBAS processing? Where is it located and why did you select this one?

Results section.

In Figure 5 and section 3.1 you talk about “ground references”. What do you mean by “ground references”? It is not clear at all. References of what? It is confusing and is not explained. You should show the mean LOS velocity of the full processed crop. There are only very few points. You should indicate that the results are in the LOS direction in Figure 5. Use a color bar with more sections. It is difficult to know where 5, 10, 15… mm/yr are located. In Figure 5 there are some points named G1, G2, G3, and G4. What do they mean?

Line 193. Do you mean +/-20 mm/yr. You should add a histogram of the velocity values. In lines 94-95 you mix the “units” mm/yr vs. mm. It has no sense. Why do you give two and four decimal parts of one millimeter? Are you so accurate? Line 192. How did you calculate these percentages? You talk about a PS-InSAR processing but the method is not explained in the methodology and the results of the velocity are not shown. It is very difficult to see the velocity map from Figure 5. There are apparently many red points (ground references) that I do not understand. If you apply PS-InSAR and SBAS, show the results of both velocity maps. It is very important to indicate in the Figures the reference used for the processing.

You should show with some Figures some examples of areas where a hazard is identified with SBAS.

Figure 7 is confusing. I do not understand what “a reference correctly detected by InSAR” is. As said before, what is a reference? Why so many references? It is not indicated in the text.

In Figure 7 you talk about FPs. What are FPs? They are not indicated in the legend. The Figure is confusing as I do not understand why the legend is present if you are not representing the velocity.

In Figure 7 you talk about InSAR, but PS-InSAR or SBAS? The paper is confusing and the results and methodology are not well described.

I do not understand line 20.

In lines 239-240 it is said that “As shown in Fig. 9, there are 327 hazards were detected by Yolo, and 35.75% of them matches the ground references”. Apart from the grammar, you should indicate some examples of the hazards detected with the Yolo algorithm. How the algorithm can identify the different types of hazards? Which is the size of the pixels in the classification? Again, I do not understand what you mean by “references”. How are they connected with the SBAS results? How many images did you use in the classification? All of this is not clear and it is difficult to assess the reliability of the classification in these mountainous areas because the different types of hazards are not clearly distinguished from the optical images.

Figure 9’s caption is confusing.

The combination of SBAS and the Yolo model is not well described in the text. How did you combine both methods? Figure 10 and their description is not clear. How can you assess the given percentages? How did you calculate them? Where is located the Nujiang valley? The results are not well described. Why do you select different areas in the figures (a, b, c…) if you do not explain them?

There are two figures 10. The second figure 10 gives some examples of the SBAS-InSAR and Yolo results but you should explain what they mean. The figure is totally understandable. Why this color bar? Use a symmetrical scale with coherent values as in previous figures. It is not possible to understand and distinguish between the velocity values. All are “green”.

Discussion. Figure 12 is not well described. Figure 12a does not represent any height and you discuss values of 2000, 3500 m… I do not understand it. You should explain the content of the axes. There is no Figure 13.

You discuss the limitation of the short dataset. As said before, why did not you use more years? You also discuss the possibility of using the DS-InSAR technique. SBAS is a DS-InSAR technique!

You also discuss the limitation of optical images for detecting hazards with ML algorithms, so because of that, I do not see any improvement in the combination of both InSAR and optical techniques apart from percentages that are not justified.

Author Response

Please see the attched file.

Reviewer 2 Report

Dear authors,

thank you very much for submitting your paper, which reports about an interesting application of “geological hazards” by means of a novel strategy which combines SBAS-InSar and the Yolo model.

Here in the following my recommendations and comments to your paper.

Line

Comment

17-18

Not clear the meaning of what you call “geological hazards”, and the deforming ones. Do you mean landslides? Please use precise and scientific words.

23-26

Not clear. Especially in the last part of the sentence. Write better please

40

Again geological hazards. Please write more scientifically

44

Manner is a not very scientific word. Try strategy or something similar

44-45

Actually, the surface deformation, is, or could be the final result of a landslide reactivation. However, surface deformation may be due also to other phenomena, such as, ground surface subsidence, or even some very local phenomena. How you discretize this in your strategy? If this is not the case, state clearly the limitation of this not direct monitoring strategy.

47

Small site? Maybe it is better site specific?

53, 54, 56, 57

It is not very elegant to write “literature” and the to cite the references. It is better to say directly “Guo et al. in 2015 [5], reported….”

63, 73

“and so on” is not scientifically sound

80

As well as demerits, maybe it is better to use “weaknesses”

82-84

Poor English, please write better

99

Are easy to occur, not very well written.. are likely to occur maybe?

Figure 1

Caption to be written again. What is the red area? Not written. Where is the LOS?

103

Why only ascending data?

120-125

Not clear. Write better please. What does it mean to remove the terrain phase in the InSAR processing? Make a reference to a paper, if you do not want to discuss it. What are the environmental factors? The InSAR target points are the scatter points?

What do you mean by landslides, and collapses?? Those are terms of vey general meaning. In the literature have been reported landslide classifications, which I suggest you to cite on this part, since each term has a very well specific meaning.

-          Cruden & Varnes, 1996: Landslide types and processes;

-          Hungr, Leroueil and Picarelli, 2014: The Varnes classification of landslide types, an update. https://doi.org/10.1007/s10346-013-0436-y

-          Cotecchia, Santaloia & Tagarelli, 2020: Towards a Geo-Hydro-Mechanical characterization of landslide classes: Preliminary results. https://doi.org/10.3390/app10227960

137

DEM errors?

Figure 5

Why only annual velocity? Could be interesting also to see the velocity on a shorter time span, e.g. monthly. Make a larger figure please. What are G1, G3 etc.? please pay attention to the captions, those are important to help the reader to understand the figures. Not really clear is red points are indicative of a increasing or decreasing distance from the satellite. State it clearly both in the text and in the caption.

143

Again terrain phase is not clear its meaning.

145-146

How and why the use of these thresholds?

216-218

How you say so? I do not see this from figure 7. Again, please help the reader in looking t the figures.

210

Why potential hazards? Those have been already occurred.

The Yolo model has been already published in the literature? Please give some references on this.

Figure 8

Do you think is it necessary to report this figure and this concept?

239

How you justify this 65.75%. please comment on this.

250-251

Poor English. Write better this caption. What do you mean by detected hazards? Those should be, according to me, displacements already occurred at the ground level. So why hazards.

Consider also that, Risk, Hazard, Vulnerability and Exposition, all have a very specific and precise meaning. There is the risk theory which explains all this. Please use the correct words in a scientific way.

Figure 10

There are 2 figure 10! Please pay attention to these issues. It is not fair to submit a paper with such issues!

However, the second figure 10 is not very clear. Make it bigger, it is not possible to read it. Improve the caption.

282

Again, potential hazard, please refer to comment to lines 250-251.

Figure 12

What does it mean what is written in the caption. It is not clear. Write better

290-291

How you say so? I think it is a too simplistic way to comment.

308-310

How you say so? Maybe it is only that you could do a more efficient and precise monitoring of the displacements, since there are more scattering points where an infrastructure exists.??

Moreover, also consider that probably the road or whatever, already existed before the starting of the monitoring period. As such, how an infrastructure may cause what you observe? Probably it is still a landslide mechanism which interacts with the man made structure.. and not the structure itself to have caused the displacement you monitor.

321

Able not bale.

322

Temporarily inactive hazards are called quiescent landslide (Cruden & Varnes 1996)

343

Again manner. Please use ways, strategies etc…

348

Following chains in high mountains.?? Not clear its meaning.

348-358

Where was all this in your study? Not sure that is what you already achieved. Maybe it is a future perspective of the work? If so, please state it clearly. I am saying this with particular reference to the second and third part of this discussion.

Author Response

Please see the attched file

Reviewer 3 Report

The manuscript entitled “Identification geological hazards in mountainous area with the combination of the SBAS-InSAR and Yolo model” is of an interesting topic. The authors used SBAS-InSAR technology to detect geological hazards in the high mountain areas along the Yunnan Myanmar border, utilizing time series SAR images from 2019 to 2020. In addition, the Yolo deep learning model was used to identify geological hazards from Gaofen-2 images. their attempt showed that the combination of SBAS-InSAR and Yolo model achieved better performance than each of them, and the coverage of hazards from the combined approach is wider than that of ground inventory. The topic of the paper is within the scope of the journal “Sensors”. I am not a native speaker but I think that extensive editing of the English language is necessary!

The positive point of the paper is the combination of two different approaches to identify and detect areas that are being affected by hazardous processes. However, the paper has some weak point regarding its structure. No information about the geology and tectonics of the area is given even if the paper deals with geological hazards. I think that the short SBAS-InSAR observation period of only one year is another weak point. Moreover, I missed the validation of the results of the applied methodology.

Here are my comments and suggestions regarding the paper:

I propose the change of the title from “Identification geological hazards in mountainous area with the combination of the SBAS-InSAR and Yolo model” to “Identification of the geological hazards in mountainous areas with the combination of the SBAS-InSAR and Yolo model

lines 17-18: what are the deforming geological hazards? Please clarify!

The authors should expand the “Introduction” section with a more detailed literature review regarding the application of the techniques they used in other case studies. They should also mention what they mean by “geological hazards”. Which specific processes are involved in this term? Landslides? Mudflows? Debris flows? Mass wasting events in general? Earthquakes?

The authors should reorganize the structure of the paper. “Study area” should be a separated section after the “Introduction” while “Data” and “Methods” should become “Data and Methods”.

Since the topic of the paper is geological hazards, the “Study Area” section should be enriched providing information regarding the geology, the geotectonic setting and the geomorphology of the area! They should also provide specific information. e.g. what is the rainfall amount? They should also provide a geological – geotectonic map of the area.

Figure 1 is too bad! An inset map showing the study area within the country should be added. It would be appreciated by readers who are not familiar with this area.

Regarding the methodology, I am wondering if an SBAS-InSAR observation period of only one year is enough to draw conclusions regarding the areas that are threatened by “geological hazards”.

Since the authors compare categorize their results according to the land use (or land cover) of the points, I think that a land use or land cover map of the study area is necessary.

line 281: I am not sure if the word anthropological factors is correct maybe the authors should change it to anthropogenic factors.

At the “Discussion” section I think a short paragraph discussing the comparison among the detected hazardous areas and the geology of the area is required since the study focuses on geological hazards.

I wonder if the authors validated the results of their approach in the field. I mean if they visited some of the detected areas and if they concluded that these are areas that need special attention. I understand that they used google earth? or satellite images but validation through fieldwork would add value to their methodological approach!

Author Response

Please see the attched file. 

Reviewer 4 Report

Dear authors, Thank you for your manuscript. Among other thinks I have left with a lot of doubts after reading it.  Firstly, everywhere you are mentioning that the two methods are combined, but is not clear what exactly and how is combined to obtain the presented as improved results... Can it be just the outputs are added to each other and compared to the reference? Which in this case i do not think it is optimal solution.  The data description is really vague and actually it is not clear how you are using different datasets without commenting their time acquisition/production component, what i mean, is that is not clear from when are the images of the optical sensor, when the reference dataset was acquired/produced... And then compared with something with known time... Because if they differ in time, they are not comparable, usable with the rest... Your conclusions for sbas-insar - that cannot detect stable landslide are, in my opinion, at the nature of the approach, similarly can be said for using one static optical image cannot measure the surface displacement... The validation approach and description is not really clear. In addition, in my opinion, the literature review is very vague, it is mainly listing works. Datasets, not sufficient. English and grammar corrections are needed.

Author Response

Please see the attched file. 

Round 2

Reviewer 1 Report

Dear authors, I am satisfied with your responses to my comments and the revision made to the manuscript.

Author Response

Thanks very much for your suggestions. 

Reviewer 2 Report

Dear Authors,

the paper has been very much improved from its first version. I think it is now ready to be published.

Kind regards 

Author Response

Thanks very much for your suggestions. 

Reviewer 3 Report

The revised version is much better than the initial manuscript! The authors followed my suggestions.

Author Response

Thanks very much for your suggestions. 

Reviewer 4 Report

Dear Authors,

I do not see major improvements following the previous review and i think the mentioned issues are not covered. The level of ambiguousness is still high.

In fact I did have specific comments in my first review to which they authors responded in the letter (even if it wasn’t really clear) but the actual answers and clarifications were not implemented in the manuscript. To be clear I will explain again my main issues with this manuscript:

  1. The dataset description was vague (even if after the review they added a bit more information) but most important, for me, was that they didn’t provide any information of data time acquisition. They are using different types of datasets in a different manner, where it is important that those datasets are coherent (between themselves) in the time. Moreover, it wasn’t clear when the reference data was collected. If the datasets are collected at different time, their work in my opinion doesn’t have a solid ground, on the contrary, if they are acquired at the same time makes their case and work better.
  2. As I explained in the first review, it is not clear how do they combine two different approaches and how do they reach the final results. Even in the authors’ response they are saying “by combining their results” … In my opinion this “combination” doesn’t bring high level of scientific soundness and actually it is not clear what did they do… What is the method that they applied behind the term “combination”?

Regards.

Author Response

Response to Reviewer 4 Comments

Point 1: The dataset description was vague (even if after the review they added a bit more information) but most important, for me, was that they didn’t provide any information of data time acquisition. They are using different types of datasets in a different manner, where it is important that those datasets are coherent (between themselves) in the time. Moreover, it wasn’t clear when the reference data was collected. If the datasets are collected at different time, their work in my opinion doesn’t have a solid ground, on the contrary, if they are acquired at the same time makes their case and work better.

Response 1: Thanks for your comments. The paper is the outcome of a project of Yunnan Province, where the remote sensing images and references were collacted at the same time (e.g., 2019-2020), by the Yunnan Institute of Geological Science. In the paper the time of Sentinel-1A acquisition is shown in Fig.3: The baseline map of Sentinel-1 images, where the x-axis is the specific acquisition date for images. The Gaofen-2 images was captured on December 5, 2020.

Point 2: As I explained in the first review, it is not clear how do they combine two different approaches and how do they reach the final results. Even in the authors’ response they are saying “by combining their results” … In my opinion this “combination” doesn’t bring high level of scientific soundness and actually it is not clear what did they do… What is the method that they applied behind the term “combination”?

Response 2: Thanks very much for your comments. As the tests indicate the SBAS-InSAR just detect moving landslides, while Yolo could find those covered by bare soil, we tried to combined the results of the two methods to enlarge the set of possible landslides. Experiments proved the effectiveness of the combination that more references are checked with the enlarged set, and both possible moving and quiescent landslides could be found for further checking or analysis. Considering the principle, characteristics, and processing techniques of the two data (SAR and optical image) are very different, and they both have evident limitations on landslide detection, new uncertainties are inevitable the proves of combining or fusing them, and will seriously influence the detection results. In our opinion, it is hard to say which method is better to fuse them, combining the results like us or inserting the SBAS-inSAR result into Yolo models, etc. as the uncertainties and related influences is very difficult to evaluate. As a first attempt to combine them in a practical landslide investigation project of the mountainous province, we adopted a direct way to do it and achieved satisfied results. We believe it is a valuable and helpful attempt to discover more possible landslides there. In the further study, we will focus on comparing different fusing methods and evaluating the uncertainties and their influences in the fusing process.